# Predictors of Pro-Environmental Intention and Behavior: A Perspective of Stimulus–Organism–Response Theory

**Agus Sugiarto** [1], **Cheng-Wen Lee** [2], **Andrian Dolfriandra Huruta** [3], **Christine Dewi** [4] **and Abbott Po Shun Chen** [5,*]

1   Department of Management, Satya Wacana Christian University, 52-60 Diponegoro Rd, Salatiga 50711, Indonesia
2   Department of International Business, Chung Yuan Christian University, 200 Zhong Bei Rd, Taoyuan City 32023, Taiwan
3   Department of Economics, Satya Wacana Christian University, 52-60 Diponegoro Rd, Salatiga 50711, Indonesia
4   Department of Information Technology, Satya Wacana Christian University, 52-60 Diponegoro Rd, Salatiga 50711, Indonesia
5   Department of Marketing and Logistics Management, Chaoyang University of Technology, 168 Jifeng E. Rd, Taichung 413310, Taiwan
*   Correspondence: chprosen@gm.cyut.edu.tw

**Abstract:** Considering the importance of pro-environmental issues, this study aims to determine the impact of level of employee education and pro-environmental intention on pro-environmental behavior based on stimulus–organism–response theory. There was a total of 425 respondents participating in the survey. This study was conducted during the period of March–June 2022. The data were analyzed using partial least square–structural equation modelling (PLS-SEM). The results indicate that the level of employee education has a positive impact on pro-environmental intention. In addition, pro-environmental intention has a significant impact on pro-environment behavior. Overall, this study sheds light on stimulus–organism–response theory in the context of Indonesia.

**Keywords:** stimulus–organism–response theory; pro-environmental; intention; behavior; education; PLS-SEM

## 1. Introduction

The environmental development pillars include six Sustainable Development Goals (SDGs). This means that the attention and commitment to environmental preservation is the focus of the international community. Issues of climate change are the trigger for this commitment. Many countries have committed to taking steps to reduce the impacts of climate change, and they have different ways of dealing with them. Factors related to pro-environmental behavior in different countries and cultures are still in question [1]. Environmental development efforts carried out with a commitment to environmental preservation can be realized in various ways supported by all stakeholders of the international community in various countries. One of the employees' efforts in achieving the SDGs is to have pro-environmental behavior that focuses on environmental sustainability in the workplace. This pro-environmental behavior is defined as all possible actions aimed at avoiding harm and/or protecting the environment [2]. In Indonesia, environmental issues (including waste) are strongly influenced by population growth, economic growth, and changes in community consumption patterns. Garbage and waste have also magnified environmental and health issues. In general, river water in Indonesia is highly polluted. In 2016 to 2017, garbage also contributed to the increasing frequency of flooding incidents in Indonesia [3]. A person who engages in positive pro-environmental behavior depends on his/her willingness and caution. Currently, there are many positive environmental activities that can be practiced at home. Separating and sorting waste types, avoiding

the wasting of water and the use of plastic, and planting trees are examples of common practices inside and outside the home [4]. A study of pro-environmental behavior revealed that people's general environmental considerations were positively related to recycling and environmental activities but not to fuel-efficient driving and sustainable use of transportation [5]. Further, several studies on pro-environmental behavior have also been carried out in organizations. The success of every effort made by an organization includes the tendency and participation of employees in performing pro-environmental behavior in the workplace. Previous research has also shown that human resource management can influence employee attitude and behavior. Therefore, sustainable environmental practices practiced by human resource management can be realized with the help of employees who have pro-environmental behavior.

In studying various predictors of pro-environmental behavior, this present study uses a theoretical model based on stimulus–organism–response theory (SOR). The simplest level of interaction occurs when someone takes an action and is given a response by another person. The term SR is imprecise because of the organism's intervention between stimulus and response. Therefore, the term SOR (stimulus–organism–response) is employed. SOR theory assumes that organisms produce behavior if there are certain stimulus conditions as well. The effect that arises is a special reaction to a special stimulus, so one can expect compatibility between the message and the communicant's reaction. Thus, the elements in this model are message (stimuli), communicate (organism), and effect (response). In this case, the message can be interpreted as any information obtained by individuals through educational processes and experiences experienced by someone. The message will be communicated at the affective level, which will react in the form of an attitude that is represented in the intention to take an action. Furthermore, these intentions will have an impact on a person's behavior or actions [6].

There have been many previous research studies on pro-environmental behavior. Many were mostly conducted in the context of consumers' pro-environmental behaviors or in the field of product and service marketing in organizations [7–11]. In addition, research on the pro-environmental behavior have also been carried out in universities [12,13]. Another study examining predictors of pro-environmental behavior found that both individual characteristics and organizational efforts affected the employees' pro-environmental behaviors. However, the effect might vary according to the types of behavior [14]. In addition, a research study found that eco-centric values, beliefs, and consciousness were predictors of pro-environmental behavior in the workplace [15]. Moreover, Farooq et al. conducted a study identifying several factors (environmental attitudes, feedback, green self-efficacy, leadership roles, organizational culture, and employee empowerment) and strategies (incentives; top management support; creating environmental knowledge and awareness; rules and regulations; and supporting sustainability) to promote ecological behavior in the workplace [13]. Another research study was also conducted to identify and quantitatively assess the importance of psychosocial and organizational factors that influenced the employees' intentions to engage in pro-environmental behavior in the workplace [16]. One of the individual characteristics related to employee intentions in pro-environmental behavior that has not been widely studied is the level of employee education. Education refers to "leading out" activity. Any experience that has a formative effect on the way people think, feel, or act can be considered educational. Generally, education is divided into stages such as preschool, elementary school, junior high school, high school, and then college, university, or internship. Moreover, there are research recommendations mentioning that research on the pro-environmental behavior of employees is very minimal [13]. Researchers have called for more research due to the limited literature [17]. Recent years have seen an increased discussion of employees' pro-environmental intentions and behaviors from various aspects. However, this paper considers the educational level of employees as an exposure variable of the existing structure of stimulus–organism–response theory (SOR). Given these considerations, this study is certainly welcome. This study also contributes to the advancement of knowledge in relation to the predictors of pro-environmental intention

and behavior. It could also help governments to make decisions on environmental issues. Moreover, it highlights the direction that a government or policymaker can take to pursue environmental intention, environmental behavior, and its education.

This study is divided into six parts. The first part explains the introduction. The second part presents the literature review and hypothesis development. The third part describes the research methodology. The fourth part presents the empirical results of this study. The fifth part elaborates the discussion. The last part highlights the conclusions of this study, including limitations and suggestions for further research.

## 2. Literature Review

### 2.1. Theory of Stimulus–Organism–Response (SOR)

Mehrabian and Russell proposed this model within the field of environmental psychology [18]. The SOR model consists of three factors: stimulus, organism, and response. This model can see the consequences of the event. Stimulation is an external force that influences individual psychological states [19]. Stimulation is also defined as the influence that stimulates the individual [20]. This model conceptualizes behavior, suggesting that a stimulus can affect the individual's physical and psychological levels in the environment that comprises the stimulus, as well as influencing the consumer's cognitive and emotional processes, which ultimately lead to a behavioral response [18]. Organisms and responses refer to the user's emotional and cognitive state, as well as the entire process of intervention between stimuli and individual responses. Stimulus and behavior are not directly causal, including affective and cognitive variables [21].

Based on stimulus–organism–response (SOR) theory, this study attempts to answer the following research questions. How do the experience and educational levels of employees in an organization relate to the intention to behave in a green manner in the workplace? We assume that a stimulus (i.e., environmental information through formal education) influences employees' internal psychological states (i.e., intention to behave in an environmentally friendly manner), which in turn motivates them to behave in an environmentally friendly manner at work. The stimulus sector, which usually attracts the most attention, represents our present awareness, or "awareness of consciousness" [22]. This sector includes the individual's active motives, moods, perceptions, cognitions, and so on [23]. A person's awareness, motives, perceptions, and cognition are certainly formed from their educational experiences, including formal educational experiences.

This study offers several key contributions to the understanding of the green behavior phenomenon. First, this research explores the factors that motivate employees' green behaviors in the workplace, thus broadening and advancing green behavior research at the individual level. Second, while SOR theory has been widely used to study other phenomena, this research will contribute to this theory by adapting it (and thereby expanding its application) to explain pro-environmental behavior. In supporting this theoretical contribution, we frame SOR theory by including one intervention variable, namely, the level of education for the context of environmentally friendly behavior. Finally, our research is expected to offer rich insights to help policy makers promote employee green behavior in the workplace.

Attitude consists of three components, which include a cognitive component, an effective or emotional component, and a behavioral component. The cognitive component of attitude refers to the beliefs, thoughts, and attributes that we would associate with an object. Basically, the cognitive component is based on information or knowledge. It is an opinion or belief segment of an attitude. It refers to the part of attitude related to one's general knowledge. Fishbein and Ajzen revealed that belief is the information a person has about an object—information that specifically relates an object and its attributes [24]. The cognitive component is the storage area where individuals organize information. A person's information is obtained through the education process. Thus, one's education will form one's cognitive component.

The second component is affective. An individual's attitude towards an object cannot be determined only by identifying their beliefs about the object because emotions work simultaneously with cognitive processes about the attitude object. Influence flows (feelings and emotions) and attitudes (evaluative judgments based on brand beliefs) are combined to propose an integrated attitude and choice model [25]. The behavioral component reflects how attitudes affect the way someone acts or behaves. This is helpful in understanding their complexity and the potential relationship between attitudes and behaviors. The behavioral component is a verbal or overt (nonverbal) behavioral tendency by an individual and consists of observable actions or responses that are the result of an attitude object [26].

This intention is the beginning of the formation of a person's behavior. The theory of stimulus–organism–response (SOR) is relevant to describe any behavior that requires planning, including the pro-environmental behavior of employees in their workplace. In the theory of stimulus–organism–response (SOR), one of the predictors of forming a person's behavior is the interest of that person to behave. The behavioral intention refers to a person's desire (interest) to perform a certain behavior. A person will perform a behavior if he/she has the desire or interest to do so. It is a function of subjective attitudes and norms towards that behavior [27]. Moreover, attitude refers to how strongly a person holds an attitude towards an action, and subjective norms become social norms associated with the action. However, the subjective attitudes and norms are unlikely to have equal weight in predicting a behavior. Depending on the individual and the situation, these factors may have a different impact on the behavioral intention, so the weight is associated with each of these factors [28], while the interest itself is influenced by many factors, and one of them is the educational factor.

The behavior can also be defined as a series of actions made by an individual, organism, system, or artificial entity in relation to itself or its environment [29]. It is the computed response of a system or organism to various stimuli or inputs—whether internal or external, and done consciously or unconsciously, overtly or covertly, and voluntarily or involuntarily [30]. Furthermore, the behavior is also a set of actions of a person in responding to something and then becoming a habit because of the values believed. Furthermore, Goleman et al. explained that a pro-environmental behavior is a human behavior in protecting and maintaining the environment in their immediate environment [31]. The interesting thing about it in human relations with the surrounding environment is the place identity and environmental awareness.

Previous research has used a stimulus–organism–response (SOR) theory perspective to stud behavior in various fields, including a SOR perspective in behavior in the field of education and learning. Among the research is that which examines what factors stimulate and influence the continuity (individual response) of students' mobile learning (M-learning). This research provides a new lens for M-learning through SOR theory [32]. In addition, there is research on the mediation of the role of emotions and experiences in the stimulus–organism–response framework in higher education [33].

The SOR perspective has also been used to examine consumer behavior, including "Black Friday Shopping Behavior among Generation Y Consumers in Botswana: Application of Stimulus–Organism–Response Theory" [34]. In addition, research has also been conducted exploring consumer behavior in virtual reality tourism using an extended stimulus–organism–response model [35]. Several other studies using the SOR perspective on consumer behavior have also been conducted by previous researchers [36–38].

Moreover, several studies on employee behavior, namely, energy-saving behavior and behavior in using transportation equipment by employees from an SOR perspective, have also been carried out [39,40]. However, the two studies on employee behavior are specific to employee behavior. Therefore, this research examines more comprehensive environmentally friendly behavior among employees in the workplace with an SOR perspective with the aspect of education as a predictor, and environmentally friendly intentions as a mediator.

Based on this argument, we examine the relationship between employee education and employee intentions in pro-environmental behavior, as well as the implications for pro-environment behavior in their workplace through the theory of stimulus–organism–response (SOR).

*2.2. Hypothesis Development*

Education is believed to be one of the predictors shaping one's behavioral intentions. In the context of consumer behavior, one study presented important findings in the field of education and purchasing intention and the relationship between education and changes in green purchasing preferences [41]. In addition, Hu and Zhang showed that the level of education and discipline had a significant effect on a person's behavioral intentions [42]. Moreover, Shimul et al. stated that through relevant environmental information or knowledge, consumers would be more educated, aiming to influence attitude in a positive way as well as purchasing intentions [11]. In perceptive SOR theory, the stimulus sector can be a person's awareness [22]. This sector includes the individual's active motives, moods, perceptions, cognitions, and so on [23]. A person's awareness, motives, perceptions, and cognition are certainly formed from their educational experiences, including formal educational experiences. Therefore, the first hypothesis that can be proposed is as follows:

**Hypothesis 1 (H1):** *The level of employee education has a significant impact on the pro-environmental intention.*

The behavioral intentions are believed to be directly related to a person's behavior. Khalid et al. found that the employees' green attitude, green subjective norms, and perceived green behavior control positively influenced the employees' required and voluntary green behaviors indirectly through their green behavioral intentions [43]. Further, the perceived organizational support for the environment reinforced the positive effect of employees' green behavioral intentions on their required and voluntary green behavior. Other studies showed that individual Green-IT attitudes and intentions had a major impact on the environment as a social behavior. As a result, the positive and essential attitude of the social sector was the main tool for efficient Green-IT implementation [44]. Similar results were also found in [11], namely, that a multicultural perspective on the relationship between a new set of cognitive and emotional factors, green customer advocacy, and feedback behavior can directly and indirectly influenced the green buying behavior. The results of other studies also confirmed that a person's attitude and intentions influenced their behavior [7]. Further, a study by Liao et al. also revealed that the behavioral intention had a significant positive effect on the choice to purchase energy-efficient equipment [9]. Therefore, the second hypothesis that can be proposed is as follows:

**Hypothesis 2 (H2):** *Pro-environmental intention has a significant effect on pro-environmental behavior.*

In addition, education is also believed to be a determinant of one's behavior. Similar with the formation of employee pro-environmental behavior, the education process also plays an important role. A research result showed that the education was such an important factor for sustainability and moreover that educated women gave the greatest value to being green while being socially minded [45]. Furthermore, similar research results supported the idea that education causes individuals to become more concerned with social welfare and therefore behaved in more pro-environmental ways [46]. An educational process was also believed to be associated with pro-environmental behavior. A study suggested that environmental education interventions could be enriched by including consensus estimates for pro-environmental intention in the assessment procedure [47]. In the study, the university's conservation-related environmental education intervention was designed to reduce errors in the pro-environmental intentions, especially their errors in predicting the pro-environmental intention. Before and after the course, the researchers

measured two intentions regarding the willingness to donate money and volunteer work for environmental causes.

The relationship between education and pro-environmental behavior was also revealed in another study that found that the educational achievement was associated with higher levels of pro-environmental attitudes and behaviors, and this estimate was strong for multiple resilience tests. Further analysis revealed that the acquisition of environmental knowledge is the channel that drives the effect of education on pro-environmental attitudes and behaviors. Next, Park and Sohn argued that knowledge was a strong factor in facilitating green buying behavior, and that the education and ongoing publicity should be designed to increase subjective knowledge as well as objective knowledge to be effective in promoting consumer green attitudes and behaviors [48]. Escario et al. also revealed that highly educated people were more involved in pro-environmental behavior [49]. Therefore, the third hypothesis that can be proposed is as follows:

**Hypothesis 3 (H3):** *The level of employee education has a significant impact on pro-environmental behavior.*

The research framework of this study can be seen in Figure 1.

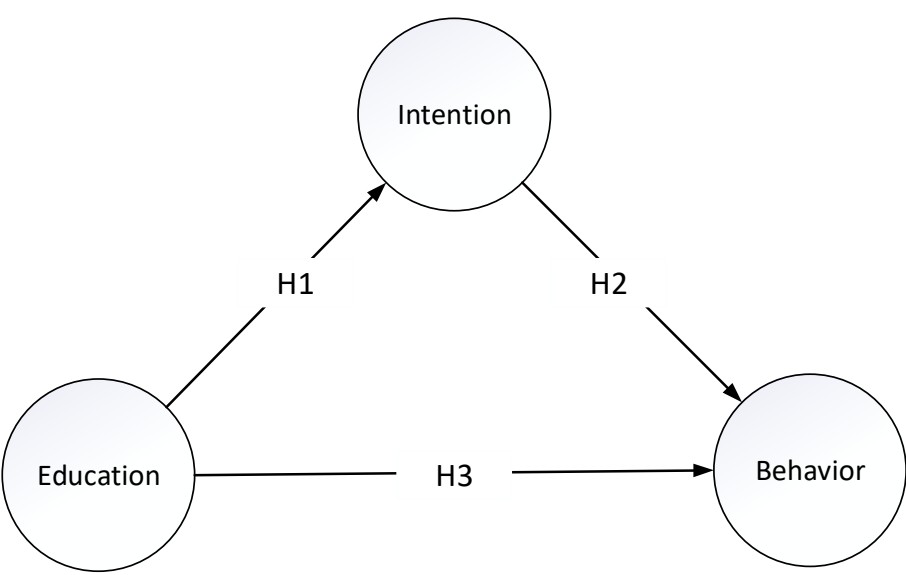

**Figure 1.** Research framework.

### 3. Research Methodology

The population of this study included employees from both private and governmental organizations in Indonesia. This study used a non-probability purposive sampling technique. In order to confirm the observation's sample size, statistical power and pointing arrows were applied. The most critical elements affecting sample size are statistical power and pointing arrows [50]. Fifty-nine samples comprised the bare minimum sample size with 80% statistical power, three pointing arrows, and $R^2$ of at least 0.25, and a probability of error of 5%. The data were collected using a questionnaire distributed online through Google Forms during the period of March to June 2022. The Google Forms were distributed via several social media (WhatsApp, Facebook, Instagram, Twitter, LinkedIn, etc.). Since more than 35 questionnaires containing the answers were either missing or considered invalid, a total of 425 samples was collected. The total number of samples used to examine the data was 425. The respondents came from several sectors, such as education (150), public administration service (130), manufacturing (78), mining (17), hospitality (35), and other (13).

For analysis purposes, this study used several constructs such as the level of employee education, pro-environmental intention, and pro-environment behavior. All of these con-

structs were evaluated using a five-point Likert scale, and the data were analyzed using PLS-SEM in the SmartPLS program. The data were examined in two stages. First, the researchers examined the reflective model consisting of internal consistency, convergent validity, and discriminant validity. Cronbach's alpha, composite reliability (CR), and rho A were the three criteria used to assess the internal consistency. A Cronbach's alpha level of more than 0.7 would be typically considered acceptable [51]. A CR value of more than 0.7 indicates that internal consistency exists [52]. A rho A value of more than 0.7 represents acceptable internal consistency [53]. Moreover, convergent validity is represented by reliability and average variance extracted (AVE). The individual indicator reliability is assessed using outer loadings. A value greater than 0.6 would be considered reliable [54]. An AVE value of more than 0.5 suggests an adequate convergence [55]. Furthermore, the discriminant validity was measured by the heterotrait/monotrait (HTMT) ratio. An HTMT value smaller than 0.85 confirms an acceptable discriminant validity [56]. This study also examined goodness of fit (GoF) index [57]. Furthermore, the researchers examined the hypotheses by using the bootstrapping technique [54].

Each scale indicator was measured on a Likert scale. Each scale was subdivided into intervals of 0.8, as shown in Table 1.

**Table 1.** Response category.

| Scale Level | Interval | Response Category |
|:---:|:---:|:---:|
| 1 | 1.00–1.80 | Very low |
| 2 | 1.81–2.60 | Low |
| 3 | 2.61–3.40 | Medium |
| 4 | 3.41–4.20 | High |
| 5 | 4.21–5.00 | Very high |

Responses from respondents were graded on a 5-point Likert scale: 5 = Very High, 4 = High, 3 = Medium, 2 = Low, and 1 = Very Low.

To measure the variable of employee intentions in the pro-environment, we used eight indicators. The eight indicators were intention to be responsible for green [58,59], intention to do green [58–60], intention to contribute to green, intention to reduce the comfort of the workspace, intention to engage in green action [60], intention to save electricity [61], intention to save water [62], and intention to save paper [63]. Several indicators were also used to measure the pro-environmental behavior in this study. They included the use of sunlight [64,65], artificial lighting [66,67], and natural air [67,68]; reducing the use of air conditioning machines [67,69]; reducing paper use in the office [70,71]; reducing office waste [72,73]; water efficiency [70,74]; and the efficient use of office computers [67,70]. Both variables are related to stimulus–organism–response theory (SOR). Stimulus–organism–response theory (SOR) suggests that intention (intention) is a decision to behave in a desired way or a stimulus to carry out an action, whether consciously or not. This intention is the beginning of the formation of a person's behavior.

Related to Law Number 20 concerning the National Education System of the Republic of Indonesia (especially in chapter I, General Provisions, Article 1, paragraph 8), the level of education is the stage of education that is determined based on the level of development of students, the goals to be achieved, and skills developed. The law indicates that the level of formal education in Indonesia consists of basic education, secondary education, and higher education [75]. Based on these references, in this study, the levels of education used to measure the employee education variable were the secondary education level and the higher education level. The levels of education used included five levels with a Likert scale weighting: (1) Senior High School, (2) Diploma (Vocational) Education, (3) Undergraduate Education, (4) Masters Education, and (5) Doctoral Education.

In this study, the learning experience represented at the last formal education level of employees was understood as a stimulus for an employee by receiving information

and knowledge about science, including knowledge about environmental preservation. In the perspective of stimulus–organism–response (SOR) theory, employee formal education experience is interpreted as a stimulus (S), which can predict a person's intentions (O) and ultimately determine employee behavior (R).

## 4. Empirical Results

There was a total of 425 respondents participating in this study, which consisted of employees who worked at the organizations. Based on their age, most of the respondents were 30–39 years old (122 respondents or 28.7%), followed by those who were 50–59 years old (107 respondents or 25.2), 40–49 years old (106 respondents of 24.9%), 20–29 years old (78 respondents or 18.4%), and 60–69 years old (12 respondents or 2.8%). The following Table 2 presents the respondent characteristics.

**Table 2.** Respondent characteristics.

| Respondent Characteristics | Frequency | Percentage (%) |
| --- | --- | --- |
| Age | | |
| 20–29 years old | 78 | 18.4 |
| 30–39 years old | 122 | 28.7 |
| 40–49 years old | 106 | 24.9 |
| 50–59 years old | 107 | 25.2 |
| 60–69 years old | 12 | 2.8 |
| Total | 425 | 100 |
| Gender | | |
| Male | 188 | 44.2 |
| Female | 237 | 55.8 |
| Total | 425 | 100 |
| Level of Education | | |
| Senior High School | 16 | 3.8 |
| Diploma (D1–D3) | 41 | 9.6 |
| Bachelor's degree | 236 | 55.5 |
| Master's degree | 105 | 24.7 |
| Doctoral degree | 27 | 6.4 |
| Total | 425 | 100 |
| Length of Work | | |
| 1–5 years | 104 | 24.5 |
| 6–10 years | 80 | 18.8 |
| 11–15 years | 59 | 13.9 |
| 16–20 years | 67 | 15.8 |
| 21–25 years | 42 | 9.9 |
| 26–30 years | 32 | 7.5 |
| 31–35 years | 29 | 6.8 |
| 36–40 years | 12 | 2.8 |
| Total | 425 | 100 |

**Table 2.** *Cont.*

| Respondent Characteristics | Frequency | Percentage (%) |
|---|---|---|
| Institution Orientation | | |
| Business | 148 | 34.8 |
| Non-Business | 277 | 65.2 |
| Total | 425 | 100 |
| Ownership | | |
| Private | 225 | 52.9 |
| Government | 200 | 47.1 |
| Total | 425 | 100 |
| Sectors | | |
| Education | 152 | 36 |
| Public Administration Service | 130 | 31 |
| Manufacture | 78 | 18 |
| Mining | 17 | 4 |
| Hospitality | 35 | 8 |
| Other | 13 | 3 |
| Total | 425 | 100 |

Source: Authors' calculation.

Based on their gender, there were more female respondents (237 respondents or 55.8%) than male respondents (188 respondents or 44.2%). Based on their level of education, most of the respondents had bachelor's degrees (236 respondents or 55.5%), followed by master's degrees (105 respondents or 24.7%), diplomas (41 respondents or 9.6%), doctoral degrees (27 respondents or 6.4%), and some only attended senior high school (16 respondents or 3.8%). Based on their work experience, this study examined respondents who had worked for at least 1 year and a maximum of 40 years. Most of the respondents had worked for 1–5 years (104 respondents or 24.5%), while there were only 12 respondents (2.8%) who had worked for 36–40 years. In addition, most of the respondents worked in the non-business sector (277 respondents or 65.2%), while the rest worked in the business sector (148 respondents or 34.8%). Based on the ownership of the organizations they worked for, most of the respondents worked in organizations owned by the private sector (225 respondents or 52.9%), Moreover, the rest worked in governmental organizations (200 respondents or 47.1%). Based on sectors, the respondents worked in the education sector (152 respondents or 36%), public administration service (130 respondents or 31%), manufacturing industry (78 respondents or 18%), mining sector (17 respondents or 4%), hospitality industry (35 respondents or 8%), and other sectors (13 respondents or 3%).

In addition to analyzing the respondent profile, this study also provided the descriptive statistics of the indicators. The results can be seen in Table 3.

Based on Table 3, the values of all constructs were in the range of very low (1.00–1.80), low (1.81–2.60), medium (2.61–3.40), high (3.41–4.20), and very high (4.21–5.00). Based on these data, it appears that the latent construct "education" had a mean value of 3.202. This means that the level of employee education was in the medium category. Moreover, the latent construct "intention" showed a mean of 4.664, which was included in the very high category. Moreover, the latent construct "behavior" had a mean of 4.264, which was included in the very high category.

Methodically, the outer loadings between a construct and its indicators indicate that all indicators (higher than 0.6) are valid, which implies that the convergent validity is accepted. The results of outer loadings can be seen in Table 4.

**Table 3.** Descriptive statistics of latent variables and indicators.

| Statistics | Mean | Min | Max | STDEV |
|---|---|---|---|---|
| Education | 3.202 | 1 | 5 | 0.841 |
| Intention | 4.664 | 3 | 5 | 0.546 |
| Behavior | 4.264 | 2 | 5 | 0.736 |
| Z.1—Intention to be responsible for green | 4.812 | 3 | 5 | 0.447 |
| Z.2—Intention to do green | 4.704 | 2 | 5 | 0.559 |
| Z.3—Intention to contribute to green | 4.720 | 1 | 5 | 0.552 |
| Z.4—Intention to reduce the comfort of the workspace | 4.245 | 1 | 5 | 0.849 |
| Z.5—Intention to engage in green action | 4.546 | 3 | 5 | 0.635 |
| Z.6—Intention to save electricity | 4.609 | 2 | 5 | 0.612 |
| Z.7—Intention to save water | 4.614 | 2 | 5 | 0.637 |
| Z.8—Intention to save paper | 4.595 | 2 | 5 | 0.666 |
| Y.1—Use of sunlight/renewable energy | 3.826 | 1 | 5 | 1.173 |
| Y.2—Reduce lights and air conditioning | 4.099 | 1 | 5 | 1.056 |
| Y.3—Reduce paper | 3.708 | 1 | 5 | 1.237 |
| Y.4—Reducing office waste | 3.781 | 1 | 5 | 1.226 |
| Y.5—Water use efficiency | 4.327 | 1 | 5 | 0.820 |
| Y.6—Use of reusable tableware | 4.407 | 1 | 5 | 0.752 |
| Y.7—Efficient use of office computer | 4.445 | 1 | 5 | 0.747 |
| Y.8—Encouraging coworkers to be green at work | 4.447 | 1 | 5 | 0.822 |

Source: Authors' calculation.

**Table 4.** Convergent validity.

| Path | Outer Loadings | *p*-Value | AVE |
|---|---|---|---|
| Y.1 ← Behavior | 0.607 | 0.000 | |
| Y.2 ← Behavior | 0.666 | 0.000 | |
| Y.5 ← Behavior | 0.733 | 0.000 | |
| Y.6 ← Behavior | 0.830 | 0.000 | |
| Y.7 ← Behavior | 0.838 | 0.000 | 0.539 |
| Y.8 ← Behavior | 0.663 | 0.000 | |
| Y.9 ← Behavior | 0.740 | 0.000 | |
| Y.10 ← Behavior | 0.767 | 0.000 | |
| Z.1 ← Intention | 0.712 | 0.000 | |
| Z.2 ← Intention | 0.801 | 0.000 | |
| Z.3 ← Intention | 0.824 | 0.000 | |
| Z.4 ←Intention | 0.677 | 0.000 | |
| Z.5 ← Intention | 0.817 | 0.000 | 0.615 |
| Z.6 ← Intention | 0.852 | 0.000 | |
| Z.7 ← Intention | 0.820 | 0.000 | |
| Z.8 ← Intention | 0.753 | 0.000 | |

Note: Outer loadings of items Y.3 and Y.4 are well below the threshold and should have been removed, as they are limiting the extraction of variance (AVE) for convergence. Source: Authors' calculation.

Table 4 shows that the AVE value ranged between 0.539 to 0.615, which is near the recommended level of 0.5. When the AVE value is less than 0.5 but the CR is greater than 0.6, the construct's convergent validity is still appropriate [76].

Table 5 shows that all constructs met the internal consistency criteria. The Rho A, Cronbach's alpha, and CR values were all greater than 0.7.

**Table 5.** Internal consistency.

| Construct | Cronbach's Alpha | Composite Reliability | Rho $_A$ |
| --- | --- | --- | --- |
| Behavior | 0.877 | 0.903 | 0.892 |
| Education | 1.000 | 1.000 | 1.000 |
| Intention | 0.910 | 0.927 | 0.918 |

Source: Authors' calculation.

Table 6 demonstrates that the discriminant validity criteria were met. The hetero-trait/monotrait (HTMT) ratio was smaller than the threshold value of 0.85.

**Table 6.** Divergent validity.

| Path | HTMT |
| --- | --- |
| Education → Behavior | 0.076 |
| Intention → Behavior | 0.738 |
| Intention → Education | 0.114 |

Source: Authors' calculation.

Table 7 reveals that the GoF index of 0.357 indicated that the goodness of fit model was considered satisfactory, and it had considerable predictive power. Bootstrapping was used to examine the hypotheses with approximately 5000 subsamples. The results can be seen in Figure 2 and Table 8.

**Table 7.** Goodness of fit (GoF) index.

| Construct | AVE | $R^2$ | $GoF = \sqrt{AVE \times R^2}$ |
| --- | --- | --- | --- |
| Behavior | 0.539 | 0.464 | - |
| Intention | 0.615 | 0.012 | - |
| Average | 0.577 | 0.238 | 0.371 |

Source: Authors' calculation.

**Table 8.** Hypothesis testing.

| Path | β | t-stat. | Bootstrapping (95% CIBC ****) | | |
| --- | --- | --- | --- | --- | --- |
| | | | Bias | Lower (2.5%) | Upper (97.5%) |
| Education → Behavior | −0.019 | 0.482 | 0.000 | −0.095 | 0.056 |
| Education → Intention | 0.110 | 2.332 ** | −0.001 | 0.019 | 0.205 |
| Intention → Behavior | 0.683 | 23.086 *** | 0.004 | 0.617 | 0.734 |

Source: Authors' calculation. ** and *** = significant at 5% and 1%, respectively. **** Bias-corrected and accelerated (BCa) bootstrap.

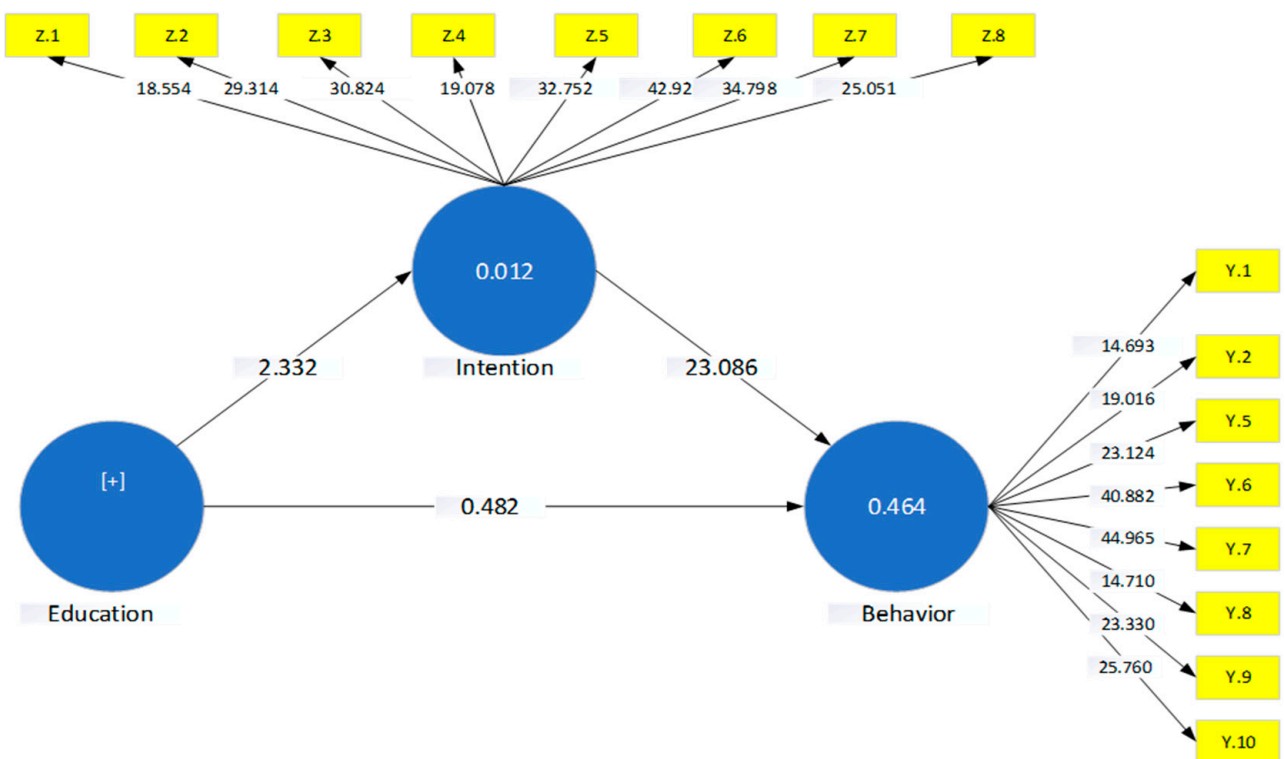

**Figure 2.** Structural output. Source: Authors' calculation.

Figure 2 and Table 8 show that the level of employee education had no significant impact on the pro-environment behavior (β3 = −0.019; $p > 0.01$; 0.05; 0.1). However, it had a positive and significant effect on the pro-environmental intention (β1 = 0.110; $p < 0.05$). Similarly, the pro-environmental intention had a positive and significant impact on the pro-environment behavior (β2 = 0.683; $p < 0.01$).

## 5. Discussion

The variable pro-environmental intention to behave in a pro-environmental manner is measured by eight indicators. The measurement of the variable shows that the indicator of interest in green responsibility has the highest level, with a mean of 4.812, while the lowest indicator is the interest in reducing their work comfort with a mean of 4.245. This finding shows that in general the research respondents have a high interest in being involved in being responsible for environmental conservation. This is quite logical because most of the respondents have relatively high levels of education, namely, a bachelor's degree (55.5%), a master's degree (24.7%), and a doctoral degree (6.4%). Thus, 86.6% of respondents have a high level of education. This is relevant to the results of research, namely, that a person's education level has an impact on the formation of their knowledge and ultimately an impact on their interest in behavior, including pro-environmental behavior. Nonetheless, indicators of interest in reducing their work comfort show the lowest results. This can also be related to the education level of respondents who have a relatively high level. In terms of the level of education, higher education tends to have a higher position and requires a more comfortable work space. This finding is also of course related to the air temperature conditions in the tropical climate of Indonesia, which has relatively hot air temperatures. Thus, the comfort of the workspace, especially setting the air temperature using an air conditioning machine, is a necessity that is difficult to avoid.

Pro-environmental behavior was measured through a number of measurement indicators, which included the use of sunlight/renewable energy, reducing lights and air conditioning, reducing paper, reducing office waste, water use efficiency, the use of reusable tableware, the efficient use of office computers, and encouraging coworkers to be green at

work [64–67,69–71,74,77]. The results of variable measurements show that all indicators are high for all categories. However, some indicators that show the lowest order are "reduce paper", "reducing office waste", and "use of sunlight/renewable energy". From this description, it appears that the respondents still have less efficient use of paper in the office than other indicators. That is why they are still less efficient in using paper at work. One of the causes is the less optimal use of paperless office technology/e-office [70,71]. This also has an impact on behavior in reducing office waste. The largest office waste is in the form of paper and plastic [72,73]. However, the behavior that still needs to be improved is the behavior in the use of sunlight/renewable energy and the urgency of improving behavior in the use of renewable energy. This finding is in line with the results of research, which show that Indonesian people still have a low tendency to use renewable energy [78], while the indicator of encouraging coworkers to be green at work is the highest indicator in the pro-environmental behavior variable.

In this study, we examined three research hypotheses. The results of the first hypothesis (H1) show that the level of employee education has a positive and significant effect on the pro-environmental intention ($\beta1 = 0.110$; $p < 0.05$). This indicated that the level of employee education and the educational process experienced by the respondents have positive and significant effects on their pro-environmental intention in their workplace. This finding certainly is consistent with the statement by Urmínová that educational level is important in green purchasing preferences [37]. In addition, the results of this study were also in line with the findings by Hu and Zhang, which showed that the level of education and discipline has a significant effect on a person's behavioral intentions [38]. Similarly, Shimul et al. found that through relevant environmental information or knowledge, consumers would be more educated, which aimed to influence positive attitudes and purchasing intentions [10].

Furthermore, the results of the second hypothesis (H2) show that the pro-environmental intention has a positive effect on pro-environmental behavior ($\beta2 = 0.683$; $p < 0.01$), which indicated that the higher the respondents' pro-environmental intentions, the more they were encouraged to perform pro-environmental behaviors in their workplace. The results of this study, especially the second hypothesis, were in line with the study by Khalid et al., which showed that the employees' green attitudes, green subjective norms, and perceived green behavioral controls positively influenced the employees' required and voluntary green behaviors indirectly through their green behavioral intentions [43]. In addition, this study supported other research that found that a person's attitude and intention influenced their behavior [7]. Furthermore, our findings were also similar with several other studies, which found that there was a positive influence of attitudes and interests on their behaviors in being pro-environmental [7,8,11,44]. This finding is in line with Park and Sohn that knowledge was such a strong factor in facilitating green purchasing behavior, and that the education and ongoing publicity should be designed to increase the subjective knowledge as well as objective knowledge to be effective in promoting the consumer green attitude and behavior [48].

In addition, the results of hypothesis (H3) show that the level of employee education is not able to directly influence the pro-environmental behavior ($\beta3 = -0.019$; $p > 0.01$; 0.05; 0.1). This finding implied that the level of employee education did not directly affect the pro-environmental behavior of employees in their workplace. This result is in contrast with another research study, which showed that educated women had the greatest value to being green while being socially minded [45]. In addition, education causes an individual to become more concerned with social welfare and therefore behave in a more pro-environmental manner [46,47,49].

In summary, the results of this study indicate that the level of employee education has a positive impact on their pro-environmental intention in the workplace. Furthermore, the pro-environmental intention also encouraged the tendency of employees to behave or act in a pro-environmental manner in their workplace. This finding supported and strengthened stimulus–organism–response (SOR) theory, which explained that the attitude

towards behavior was an important subject that could predict an action. However, it was necessary to consider a person's attitude in examining subjective norms and measuring a person's perceived behavioral control. If there was a positive attitude, the support from people around, and a perception of ease because there were no barriers to behave, the person's intention would be higher [79]. The intention was a decision to behave in a desired way or a stimulus to carry out an action, whether it was done consciously or not [80].

## 6. Conclusions

The first conclusion of this study is the level of employee education has a positive effect on the pro-environmental behavior, indicating that the level of employee education and the educational process have positive impacts on their attitude tendency to behave in a pro-environmental manner in their workplace. Second, the pro-environmental intention has a positive effect on pro-environmental behavior, indicating that the higher the respondent's interest in the pro-environmental behavior, the greater the tendency to encourage them to behave and act in a pro-environmental manner in their workplace. Third, the level of education does not directly affect the pro-environmental behavior of employees in the workplace.

### 6.1. Practical and Theoretical Implications

The practical implications of this study are that education contributes to shaping employees' pro-environmental intentions and subsequently contributes positively to pro-environmental behavior. Therefore, we suggest that management commit to integrating pro-environment themes in the employee training and education curriculum in the company. In addition, the management implication suggests that the organizers of formal education programs in Indonesia, from elementary to higher education levels, integrate pro-environment themes in their learning curriculum. The theoretical implication actually supports stimulus–organism–response theory (SOR), which explains that the attitude towards behavior is an important subject that can predict an action [32–34,36–40]. In this context, the pro-environmental intention can predict the tendency to act in a pro-environmental manner for employees in their workplace.

### 6.2. Limitation and Suggestion for Future Research

Along with its strengths, this study has a number of limitations. The role of education in the study is based on the level of education. Therefore, a future study could consider education through any training or program that enhances pro-environmental behavior. Conducting a longitudinal study that considers the effects of time series and the cross-sectional scope of pro-environmental behavior is strongly encouraged to boost research validity.

**Author Contributions:** Conceptualization, A.S.; methodology, A.D.H.; software, C.D.; validation, C.-W.L.; formal analysis, A.D.H.; investigation, C.-W.L.; resources, A.S.; data curation, A.P.S.C.; writing—original draft preparation, A.D.H.; writing—review and editing, A.S.; visualization, C.-W.L.; supervision, A.P.S.C.; project administration, C.D.; funding acquisition, A.P.S.C. All authors have read and agreed to the published version of the manuscript.

**Funding:** This research is partially supported by the National Science and Technology Council, Taiwan (Grant number: MOST-111-2637-H-324-001-) and the Ministry of Education, Taiwan (Grant number: 1110035928).

**Institutional Review Board Statement:** Not applicable.

**Informed Consent Statement:** Not applicable.

**Data Availability Statement:** Not applicable.

**Conflicts of Interest:** The authors declare no conflict of interest.

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
