# Peer review of "Predictors of Pro-Environmental Intention and Behavior: A Perspective of Stimulus–Organism–Response Theory"

_sustainability, doi:10.3390/su142316047_

Round 1
Reviewer 1 Report
Introcuction
In the introductory section appear two separated narrative lines. The first one specific to the pro-environmental intention in the workplace, and the second one general to the whole territory of Indonesia. Which line is the leading and most important to the authors? It is necessary to be specific on this issue.
The argumentation used in the study should be consistent. Extremely different argumentation tactics should be located in the appropriate context. The issue here is the functionalization of the argument and, what it is intended to be used for. Whether to demonstrate some kind of global coherence of environmental action, or to demonstrate differences. Therefore, strategies and tactics related to personal choices, such as those of Lithuanian residents, need to be considered through the prism of socio-cultural differences that reveal themselves strongly between, for example, Central and Central-Eastern and Eastern Europe and other continents.
In the introductory section, the authors cite a number of theoretical assumptions and results of previous research, but little of this context applies to the present topic. These theories and results apply to other age groups and other educational groups. Here, as I understand it, the study of a specific professional group will be crucial.
Literature Review
An important, and missing, context in this part of the study is the three-element theory of attitudes and at least elements of theory related to the psychology of persuasion and the psychology of communication, since every act of communication is always realized on 3 levels: (1) cognitive - purely informational, related to the providing of factual information about the object of communication; (2) affective, related to emotions toward the object being communicated about; and (3) volitional, related to actions taken in response to the message. These three levels correspond to the three-element structure of attitudes. As a theoretical context, they should appear in this work, since the essence of this study relates precisely to the measurement of behavior.
There are some inconsistencies and distortions in the literature section, as the authors began the discussion of indicators in the later analysis. First of all, this element should be transferred to the methodology. Second, how were these indicators selected (light, air and others?) From what pool of indicators were they selected? And why exactly was such a decision made? This element of the study is methodologically non-transparent.
Hypothesis Development
H1 - The level of education has a significant impact on pro-environmental behavior, but this factor can be overruled and marginalized by another factor related to the sector of the economy from which the worker comes. So we come to the issue of negotiation of meanings. A brilliantly educated person may be an employee of a corporation that is the biggest polluter in some territory. I advise being careful about absolutizing education without considering the broader context, e.g., without indicating the specific sector of the economy the authors intend to look at.
H2 - Caution as with hypothesis 1. I see the danger of absolutizing intentions. Rather, I would see the question here as: does intention have such an impact. Intention may again be subject to negotiation of meanings and individual, personal calculation of gains and losses.
H3 - Hypotheses formulated in this way should be better placed in a situational context and more accurately represent possible decision-making processes and the factors influencing them.
Research Methodology
Among the many different social stratification criteria used in the survey, one very important one is missing. That is an indication of the industries from which the survey participants come. Are these industries critical on the environment? Is it chemical, heavy industry? Is it the soft sphere of the economy related to services, that do not consume natural resources? Is it the public service sector? This is a necessary indication, as it translates into assumptions, hypotheses and projects the research posture.
Discussion
This part of the study reveals that the study has little informative value. The essence of informativeness is that new information dominates over old information in a significant way and contributes to expanding the possibilities of viewing a phenomenon. In the discussion, one finds mainly statements that, in connection with hypotheses H1, H2, H3, something has some influence or no influence on something else. Demonstrating influence is also revealing, what this influence consists of, in what it manifests itself. Here such information is lacking.
Conclusion
The conclusions presented do not have a significant informative value. If we use the illustrative metaphor of a medal here, the authors draw attention to only one side of the medal, completely ignoring the other. The authors show, that a phenomenon is occurring, but do not explain, what it consists of and what it manifests itself in.
Reviewer 2 Report
First, in the introduction section, is the literature basis sufficient for the identification of research gap? Currently, researchers are paying more attention to the employees’ pro-environmental behavior, so it is necessary to reorganized the existing literature and clarify the necessity and theoretical significance of this research.
Second, this study proposed that the pro-environmental intention has an mediating effect on the relationship between employee education and pro-environmental behavior. But this does not appear in the hypothesis development section, so authors should add the mediation effect hypothesis. In addition, some conclusions are inconsistent with the data results, for example, “The results of the 392 first hypothesis (H1) show that the level of employee education has a positive effect on 393 the pro-environmental behavior (β1 = 0.109; p < 0.05).” This need carefully check.
Third, there are 8 and 10 indicators of pro-environmental intention and behavior, respectively. Why choose these indicators to measure pro-environmental intention and behavior? The reasons for the choice of the indicators are not reflected in the manuscript. Thus, authors should explain the literature basis of indicators selection.
Fourth, the proposed practical implications are not sufficient, and it is necessary to further reorganized practical implication according to the discussion and conclusion.
Reviewer 3 Report
The title as research question is clear, explicit and focused on the research core.
Recommend English Proofreading which will improve the quality and clarity of the text.
Abstract: The abstract is too brief. Strong motivation of the research is necessary. It should be improved by presenting in summary each section of the paper. There is provided brief information about methodology and abstract information about results. Should be included more focused information about the research scope, utility and original contributions.
KEYWORDS – could be improved with 1, or 2 keywords referring to the specific concepts investigated.
The Introduction: presents the paper framework and connection between pro-environmental behavior and the theory of planned behavior – arguing also the necessity of this approach both for researchers and businesses.
Within the Introduction is brief as it states the motivation and paper structure:
This study is divided in six parts. The first part explains the introduction. The second part presents the literature review and hypothesis development. The third part describes the research methodology. The fourth part presents the empirical results of this study. The fifth part elaborates the discussion. The last part highlights the conclusions of this study, including limitations and suggestions for further researches.
Literature Review is basic, as to remind the fundamental theories the paper relies on: Theory of Planned Behavior; pro-environmental behavior; human relations; environmental awareness ..
Hypothesis Development – is elaborated for 3 Hypothesis throughout literature review and predictive thinking.
Research Methodology – is based on Structural Equation Modelling using (PLS-SEM).
The steps and methods of the research are clearly explained and represented in Tables and text.
Throghout the PLS-SEM methodology, the authors elaborate the mediation and validation of hypothesis in different sub-sections and provide the necessary description and argumentation.
Empirical results -are presented in detail, in tables and text. In this section, authors focus on presenting the quantitative and statistical data rather then interpretation of results.
Discussion section - comprises the comments for validating the hypothesis and describing the data obtained. It has the necessary flow in arguing the results.
Conclusions
Conclusions are formulated rather brief but summarize well-enough the results for the research. Authors insist on the results, connecting these to the global framework of the domain, both in theory and practice.
References
References do not fully correspond to the journal format and could be improved with more up-to-date items.
Reviewer 4 Report
Comments to the Author
Recommendation: Major Revision
First, I would like to thank the Editor for trusting me with the opportunity to review this research, "Predictors of Pro-Environmental Intention and Behavior: An Empirical Analysis of Private and Public Sector in Indonesia.” Although the theme of the study is interesting and seems to have the potential to contribute to the literature on pro-environmental behavior; however, the seriously lacks sound theoretical support and logical background. The most serious problem I find with the study is its hypothesized framework conceptualizing the direct role of only Education while the study itself has highlighted (in the introduction section) that a plethora of published studies is available predicting pro-environmental behavior of consumers, university students, and teachers, employees conceptualizing different theories and organizational and individual variables. Thus, only checking the direct role of Education on pro-environmental behavior does not justify. Moreover, another major flaw is the application of the theory of planned behavior (TPB), while the study does not conceptualize a single underlying preditor of the TPB. I wonder I did the authors claim to have the study based on the TPB. Based on my evaluation, I am compelled to warrant a rejection. The detailed comments are as follows:
1- The major problem with the introduction section is its inability to arguably present the research problem it looks to address. It just kept talking about what other studies researched and never stressed on what this study intends to proffer. There is no clarity on what exactly the objectives of the study are and why the study chose to cognize the direct role of Education only while the claims to have the data of private and public sector employees who’s pro-environmental behavior might be affected (favorably/adversely) by several organizational and individual factors.
2- The study has based its hypothesized model on the TPB while none of the underlying factors (viz. behavioral attitude, subjective social norms, and perceived behavioral control) has been instrumented in the study’s model.
3- There is no information on what kind of Education’s role the study is checking on pro-environmental behavior. Were the employees imparted any such education through any training or program so that their pro-environmental behavior could be triggered?
4- In the methods section, nothing has been reported on how the sample was selected and determined, companies from what specific sector/industry/regions were chosen, and how the data were collected using what sampling method.
5- Nothing has been reported on ‘from where the measurement scales were adopted to measure the latent constructs of the study. The study also reports nothing about how the data were screened for missing/unengaged responses and outliers.
6- The authors made a table for each demographic variable which seems unnecessary. All the demographic variables could have been portrayed in one single table.
7- The descriptive statistics should have been furnished only for latent constructs, not for each observed item.
8- It is evident from Table 8 that Education is a single-item variable scale/categorical/binary variable, which is very problematic when it comes to variance explanation.
9- Outer loadings of items Y.3, Y.4, and Y.8 are well below the threshold and should have been removed as they are limiting the extraction of variance (AVE) for convergence.
10- The presentation and reporting of the results also seem haphazard. The authors have presented the model from the Smart-PLS model showing t-values instead of presenting the model with standardized path coefficients bootstrapped significance levels.
11- No indirect path (Education à intention à behavior) has been hypothesized and tested, while Table 12 reports it.
12- In terms of discussing the findings and offering implications for the theory and practice, this study offers almost nothing. Discussion is all about comparing/confronting the findings of the previous studies and does not discuss the findings with implications. A new section, “Implications for the theory and practice,” needs to be added, providing the specific implications of the study’s findings for the theory and practice.
13- The study also needs moderate language editing and proofreading.

Round 2
Reviewer 1 Report
The completed and revised version of the article meets the criteria for such scientific papers. In the supplemented version, the article more clearly shows the research procedure, the study along with the results and the conclusions drawn from the study.
Reviewer 4 Report
Comments to the Author
It is appallingly disappointing to see that the authors did not put any effort into addressing the comments and improving the manuscript accordingly. In the introduction and review section, they simply clubbed several paragraphs together and highlighted them to represent the improvement which is actually falsification. Looking at the least seriousness put in by the authors in incorporating the comments/suggestions and improving the manuscript, I am compelled to warrant the rejection of the manuscript. Detailed comments are below.
1- As the major problem with the study is claiming to have it based on the TPB theory while not incorporating any of the underlying factors of TPB in the study’s model. The authors should have refrained from the use of the TPB and instead used any other theoretical framework like the Stimulus-Organism-Response Model, which doesn’t have a preset of underlying factors rather postulates that external stimulating factors (Education in the case of this study) influence organism (cognitive variables, i.e., Intention in this study) which ultimately forms response (i.e., behavior in this study). Moreover, I commented on improving the clarity of the need and objectives of the study, but the authors did nothing.
2- No clarification has been provided on the kind of education the study talks about rather, the authors copied and pasted some paragraphs from the study as a response to my suggestion/comment.
3- In the comments on the methods section, I asked the authors to report the sectors/industries/companies sampled for data collection, but the authors did not address the comment.
4- Again, the authors did not address the comment “nothing has been reported on ‘from where the measurement scales were adopted to measure the latent constructs of the study.” The study also reports nothing about “how the data were screened for missing/unengaged responses and outliers.”
5- The descriptive statistics should have been furnished only for latent constructs, not for each observed item.
6- The discussion is still too weak; instead seems weaker than before. I wonder what did the authors do with the discussion in the name of improvement. They did almost nothing but highlighting the text and reshuffling the citations. I also suggested adding an implications section, but the authors did not even take trouble incorporating it into the study. Although, in the conclusion section, the authors tried to proffer some very general implications for practice and stated that “The theoretical implication ………...predict an action,” while the study doesn’t even use a single underlying factor of the TPB and states the same as its limitation in the very next paragraph. Then I wonder how this study theoretically strengthens the TPB.

Round 3
Reviewer 4 Report
I commend the authors for putting in serious efforts to improve the manuscript. Now, the theoretical background of the manuscript, with the S-O-R model, seems robust and makes sense. The authors have also improved on the other aspects of the manuscript.